# The Associations Among Health-Promoting Lifestyle, eHealth Literacy, and Cognitive Health in Older Chinese Adults: A Cross-Sectional Study

**DOI:** 10.3390/ijerph17072263

**Published:** 2020-03-27

**Authors:** Shao-Jie Li, Yong-Tian Yin, Guang-Hui Cui, Hui-Lan Xu

**Affiliations:** 1Department of Social Medicine and Health Service Management, Xiangya School of Public Health, Central South University, Changsha 410078, China; ii10233972@csu.edu.cn; 2School of Nursing, Shandong University of Traditional Chinese Medicine, Jinan 250355, China; yinyongtian2004@163.com; 3School of Acupuncture and Tuina, Shandong University of Traditional Chinese Medicine, Jinan 250355, China; cgh1622040141@163.com

**Keywords:** health-promoting lifestyles, eHealth literacy, cognitive health, older adults

## Abstract

*Background:* Healthy lifestyles and health literacy are strongly associated with cognitive health in older adults, however, it is unclear whether this relationship can be generalized to health-promoting lifestyles and eHealth literacy. To date, no research has examined the interactive effect of health-promoting lifestyles and eHealth literacy on cognitive health. *Objective:* To examine the associations among health-promoting lifestyles, eHealth literacy, and cognitive health in older adults. *Methods:* Using a stratified cluster sampling method, we conducted a survey with older adults in four districts and two counties in Jinan (China). Older adults (*n* = 1201; age ≥ 60 years) completed our survey. We assessed health-promoting lifestyles, eHealth literacy, and cognitive health, and collected participants’ sociodemographic information. *Results:* Health-promoting lifestyles and eHealth literacy were significantly and positively associated with cognitive health (both *p* < 0.01). In addition, eHealth literacy was positively associated with health-promoting lifestyles. Moreover, the interaction of health-promoting lifestyle and eHealth literacy negatively predicted cognitive health (*β* = −0.465, *p* < 0.01). *Conclusions:* Health-promoting lifestyles and eHealth literacy were associated with the cognitive health of Chinese older adults, both independently and interactively. Further, eHealth literacy was associated with health-promoting lifestyles in older adults. Therefore, interventions regarding healthy lifestyles and eHealth literacy would benefit older adults.

## 1. Introduction

Cognitive health is an important component of brain health and it refers to the ability to clearly think, learn, and remember [1]. Cognitive impairment is one of the most common indicators of human aging. Certainly, there is no doubt that poor cognitive health, including mild cognitive impairment, dementia, and so on, affects the health status and well-being outcomes of older adults. With the intensification of global aging, poor cognitive health not only negatively influences the health of the elderly, but also increases the burden of care and psychological distress on families [2]. In one systematic review, the world-wide incidence of mild cognitive impairment per 1000 person-years is 22.5 for ages 75–79 years, 40.9 for ages 80–84 years, and 60.1 for ages 85+ years [3]. Approximately 9.5 million Chinese older adults were estimated to have dementia in 2017, accounting for 5.3% of all older adults in China [4]. Thus, maintaining and promoting cognitive health and decreasing the risk of dementia is of great significance in improving the health status of older Chinese adults.

Cognitive health is related to many factors. For example, according to the World Health Organization’s guide for caring for people with dementia, adopting a healthy lifestyle can reduce the risk of cognitive decline and dementia [5]. Health-promoting lifestyles refer to the spontaneous and multifaceted perceptions and behavior adopted by an individual to maintain his or her health promotion status and to achieve self-satisfaction and self-realization [6]. Research has shown that health-promoting lifestyles play a positive role in reducing depression [7] and improving quality of life among older adults [8], which has become an important topic in the field of prevention and control of chronic disease and health management. At present, research on the relationship between health-promoting lifestyles and cognitive health has not been reported in the literature. Instead, the extant literature is more focused on reading literacy [9], dietary patterns [10], physical activity [11], smoking [12], drinking [13], and other individual lifestyles. Health-promoting lifestyles as a comprehensive concept, including self-actualization, health responsibility, exercise, nutrition, interpersonal support, and stress management, are more meaningful than individual lifestyles. Thus, it can be inferred that individuals with high health-promoting lifestyles may have better cognitive health. Accordingly, we propose our first hypothesis:

**Hypothesis 1 (H1):** 
*Health-promoting lifestyles will be positively associated with cognitive health.*


One cohort study suggests that low health literacy was associated with cognitive decline [14]. Health literacy refers to the degree to which individuals have the capacity to obtain, process, and understand basic health information and services needed to make appropriate health decisions [15]. Research indicates that health literacy is a protective factor against cognitive decline [16,17]. However, it is not clear whether this relationship generalizes to eHealth literacy, which is an extension of the concept of health literacy [18]. With the development of the Internet, eHealth literacy has been widely used to measure an individual’s ability to apply, evaluate, and practice health information over the Internet. However, the association between eHealth literacy and cognitive health has not been extensively investigated. However, given the relationship between health literacy in general cognitive health, we infer that a relationship likely exists between eHealth literacy and cognitive health. Accordingly, we propose our second hypothesis:

**Hypothesis 2(H2):** 
*eHealth literacy will be positively associated with cognitive health.*


In general, an individual’s health is the result of a combination of many factors, some of which are interrelated [19]. There is no doubt that an interaction does exist between health-promoting lifestyles and eHealth literacy. One cross-sectional study found that health-promoting behaviors were correlated with eHealth literacy in patients with Type 2 diabetes [20]. This relationship has also been observed in college students [21]. However, it is not clear whether the relationship applies to older adults. In addition, most studies of cognitive health in older adults have only focused on the independent effects of healthy lifestyles or health literacy on cognitive health, and have not investigated the interactive effects of health-promoting lifestyles and health and eHealth literacy. Considering the rapid development of online health services, understanding the interaction of health-promoting lifestyles and eHealth literacy will contribute to enhancing and promoting advances in the development of prevention strategies for cognitive health. Thus, we propose our third and fourth hypothesis hypotheses:

**Hypothesis 3(H3):** 
*eHealth literacy will be positively associated with health-promoting lifestyles.*


**Hypothesis 4(H4):** 
*The interaction of health-promoting lifestyles and eHealth literacy will affect cognitive health.*


## 2. Methods

### 2.1. Study Participants and Procedure

The data analyzed for this study came from a survey on mild cognitive impairment and its influencing factors in older adults in Jinan city. The sample size was calculated using the following formula: $n=\frac{\mu_{\alpha /2}{}^{2}\pi \left(1-\pi \right)}{\delta^{2}}$, which is commonly used in cross-sectional studies in epidemiology [22]. In one systematic review, the prevalence of mild cognitive impairment in older Chinese adults was 14.7% [23]. Thus, π was set at 14% in the present study. Using *μ* = 1.96, d = 0.2, and α = 0.05, our estimated minimum sample size was 1156. Consequently, from January to February 2019, we adopted a stratified cluster sampling method to conduct surveys in 4 districts and 2 counties in Jinan city. We randomly selected two streets or towns in each district or county respectively, and two communities or natural villages in each street or town respectively to administer our survey. A total of 8 urban communities and 16 natural villages were selected. Inclusion criteria were as follows: age ≥ 60 years; clear consciousness; language communication and cognitive judgment ability; the ability to complete basic measures; the ability to surf the Internet; the ability to provide voluntary informed consent, and cooperated with the study. In contrast, the exclusion criteria were as follows: patients with severe and terminal diseases (as reported by family members); patients with severe cognitive impairment such as dementia, confusion, etc. (as reported by family members); patients with hearing and visual impairments or sequelae of stroke without the ability to communicate. We visited 1250 participants, but 49 participants were excluded as they did not meet our inclusion criteria or were not interested in participating. Thus, the final sample used for analysis was 1201 (participation rate = 96.1%).

Our study was conducted according to the Declaration of Helsinki and we obtained approval from the Medical Ethics Committee of Central South University (Identification code: CTXY-150002-7). All participants were assigned code identifiers and the data were stored on a password-protected computer. All participants were told the purpose and content of the study and provided informed written consent. For participants who had difficulty reading or filling in the survey, a trained college student conducted a one-on-one interview to help participants complete the survey. The survey lasted approximately 15 minutes and participants received a small gift as compensation for participating.

### 2.2. Measures

Our self-report survey was comprised of four sections: participant sociodemographic information, health-promoting lifestyle, eHealth literacy, and a mini-mental state examination. Sociodemographic information assessed participants’ age, gender, residence, education (primary school and below, junior middle school, high school, university/college and above), marital status (married, other), and family economic level (low, medium, high).

We measured health-promoting lifestyles using the Health-Promoting Lifestyle Profile (HPLP) [24]. This scale consists of six dimensions and 42 items, including self-actualization (14 items), health responsibility (nine items), exercise (three items), nutrition (five items), interpersonal support (five items), and stress management (six items). Participants rated responses on a 4-point Likert scale (1 = never, 2 = sometimes, 3 = often, and 4 = routinely). The total score ranged from 42 to 168, with greater scores demonstrating a healthier lifestyle. HPLP has been widely used to measure healthy lifestyles and has excellent psychometric properties [25]. In this study, Cronbach’s alpha was 0.96.

We measured eHealth literacy using the eHealth Literacy Scale (eHEALS), developed by Norman and Skinner [26] and translated into Chinese by Yu [27]. This scale consists of three dimensions and eight items: application ability (five items), evaluation ability (two items), and decision-making ability (one item). Each item is rated on a 5-point Likert scale ranging from 1 = strongly agree to 5 = strongly disagree. Participants’ total score ranges from 8 to 40, with higher scores indicating greater eHealth literacy. In this study, Cronbach’s alpha was 0.97.

Lastly, we measured cognitive health using the mini-mental state examination (MMSE), which was developed by Folstein, Folstein, and McHugh [28]. This scale consists of 30 items and assesses 5 domains of cognitive function, including: time and spatial orientation, attention, numeracy and memory, language ability, and recall ability. For this measure, a researcher asks participants questions, and participants then respond. Response options for each item are scored as: 0 = wrong or unable to answer or 1 = true. The total score ranges from 0 to 30, with higher scores indicating better cognition. According to prior research [29], a score less than 24 indicates mild cognitive impairment in older adults. In the present study, we used the total MMSE score as a measure of cognitive health. This scale has been successfully used in cohort studies with older Chinese adults [30]. In this study, Cronbach’s alpha was 0.91.

### 2.3. Data Analysis

We used IBM SPSS 25.0 (SPSS Inc., Chicago, IL, USA) for data entry and statistical analyses. We conducted tests of normality and homogeneity of variance. Measurement data consistent with normal distributions were expressed as the mean (*SD*), and enumeration data were expressed as *n* (%). Descriptive statistics were used to describe participants’ sociodemographic characteristics and the scores of HPLP, eHEALS, and MMSE. One-way analysis of variance (ANOVA) and *t*-tests were used to compare MMSE and HPLP scores across participant characteristics. Pearson’s *r* correlations and linear regression models were used to test hypotheses 1, 2 and 3. Multiple linear analysis was used to test hypothesis 4.

## 3. Results

### 3.1. Descriptive Statistics

Sample characteristics are presented in Table 1. The mean age of our participants was 70 years (*SD* = 6), with a range of 60-97 years. Of the 1201 participants, 53.4% were female. Most of the participants were married (73.6%) and were not highly educated. In terms of the participants’ family economic level, 10.5% of them exhibited high economic status. The mean HPLP score was 112.23 (*SD* = 23.25).

The mean eHEALS score was 17.24 (*SD* = 9.34), and the mean item score was 2.16 (*SD* = 1.17). The mean MMSE score was 27.07 (*SD* = 4.47). The prevalence of mild cognitive impairment was 16.0% (192/1201). There were significant differences in age, gender, residence, education level, marital status, and family economic level on MMSE and HPLP scores (all *p*-values < 0.05).

### 3.2. Correlations between Health-Promoting Lifestyle, eHealth Literacy, And Cognitive Health

Pearson’s *r* correlations examining health-promoting lifestyle, eHealth literacy, and cognitive health are presented in Table 2. 

### 3.3. Multivariate Line Regression Analyses

We controlled for demographic variables (i.e., age, gender, residence, education level, marital status, and family economic level), and estimated parameters for four regression models. 

As shown in Table 3, in Model 1, health-promoting lifestyles exhibited a major effect on cognitive health (*β* = 0.262, *p* < 0.001), supporting hypothesis 1. Model 2 suggested that eHealth literacy positively predicted cognitive health (*β* = 0.152, *p* < 0.001), supporting hypothesis 2. Model 3 suggested that eHealth literacy positively predicted health-promoting lifestyle (*β* = 0.381, *p* < 0.001), supporting hypothesis 3. Model 4 indicated that the interaction of health-promoting lifestyle and eHealth literacy negatively predicted cognitive health (*β* = –0.465, *p* < 0.01), indicating that there was a multiplicative interaction between eHealth literacy and health-promoting lifestyle on cognitive health.

## 4. Discussion

Cognitive decline is a serious health hazard to older adults’ physical and mental health. In the current study, the prevalence of mild cognitive impairment was 16.0%, which is higher than previous results of meta- analysis in China (14.6%) [23]. Thus,it is important to take the necessary measures to prevent and control the incidence of cognitive decline in older adults. Furthermore, we found that males obtained higher scores on the MMSE than females, higher levels of education of older adults were associated with higher MMSE scores, and married older adults obtained higher scores on the MMSE than non-married individuals. These findings are similar to those reported in other studies [29]. These results suggest that gender, education level, and marital status play important roles in cognitive health in older adults. When conducting cognitive health interventions, individual differences should be fully considered.

Our results indicated that health-promoting lifestyles were seem positively associated with cognitive health, which supports our hypotheses 1. Previous research on healthy lifestyles and cognitive health typically only discussed the influence of a particular lifestyle factor on cognitive health. For example, behavioral health risks negatively predict cognitive health [31]. In contrast, our study examined healthy lifestyle variables from multiple perspectives, enabling us to have a clearer and more comprehensive understanding of the associations among healthy lifestyles and cognitive health.

In addition, our study found that older adults with high eHealth literacy exhibited better cognitive health, which supports our second hypothesis. In prior studies, individuals with higher eHealth literacy were more likely to adopt a healthy lifestyle [32] and use health services [18]. This suggests that older adults with higher eHealth literacy are more motivated to maintain and promote their health status. These findings suggest that when such individuals come to realize they are experiencing problems with their cognitive functions, they may collect online health knowledge, actively seek help from medical staff, and take various measures to maintain their cognitive health, thereby improving their cognitive function. In addition, with the convenience of electronic devices and the Internet, people are able to surf the Internet anytime and anywhere, making it easier for individuals with high eHealth literacy to search and apply health knowledge, which is beneficial for maintaining cognitive health. The results of our eHEALS analyses showed that older adults exhibited a lower eHealth literacy level (the mean eHEALS item score was 2.16 (below the median of 3), indicating that most older adults lacked eHealth literacy. This finding illustrates to the importance of improving older adults’ eHealth literacy, which is of great significance to the promotion of cognitive health.

Model 3 indicated that eHealth literacy was also seem positively associated with health-promoting lifestyle, which supports our third hypothesis. The Knowledge-Attitude- Practice (KAP) Model [33] suggests that knowledge is the basis of behavior change, and beliefs and attitudes are the driving forces of behavior change. This can explain the impact of eHealth literacy on health-promoting lifestyles. eHealth literacy can affect the degree to which individuals use health services and access health resources. It is possible that individuals with higher eHealth literacy have greater access to resources that promote health and high awareness and belief in health maintenance, which can increase endogenous motivation of health status promotion, thereby encouraging individuals to adopt healthy lifestyles.

The present study tested whether the interaction of health-promoting lifestyles and eHealth literacy affected the cognitive health among Chinese older adults. As expected, the interactive effect of health-promoting lifestyle and eHealth literacy was significant. According to the Health Literacy Skill Model [34], older adults with high eHealth literacy may be more effective at adopting healthy behaviors and achieving healthier outcomes. This means that higher eHealth literacy and higher health-promoting lifestyle will be a cumulative effect, which would be extremely beneficial for the maintenance and promotion of cognitive health. However, because the interaction term negatively predicted cognitive health, this suggests that the predictive effect attributed to health-promoting lifestyles or eHealth literacy is decreased by improvements in the other. This is of great significance for the future research on cognitive health intervention in older adults. The study suggests that we can maintain and promote cognitive health in the older adults by improving their eHealth literacy and implementing healthy lifestyle interventionsin the future. Considering that lifestyle intervention is a long-term process, the issue of adhering to it must be considered. Thus, we can explore relevant interventions emphasizing eHealth literacy. At present, the market for online services for older adults is blossoming [35]. Such services provide new opportunities for online interventions in cognitive health. Additionally, portable devices such as smartphones are beneficial for improving intervention adherence [36]. Therefore, our study suggests that smartphone-based eHealth literacy interventions can be designed and used to improve cognitive health in older adults. Doing so could improve older adults’ health care awareness and the utilization of health services and stimulate individuals’ internal motivation to maintain good cognitive function.

Nevertheless, the present study has several noteworthy limitations. First, our study was cross-sectional, thus it cannot identify causal relationships. It is possible that cognitive health may in turn affect eHealth literacy and health-promoting lifestyles. Given this, in the future, longitudinal studies should be conducted to determine causal relationships among interventions and outcomes and the mechanisms by which they are effective. Second, our interaction analysis is only a statistical interaction, and whether it has practical public health significance remains to be further studied. Third, the developers of eHEALS indicated that this scale has certain defects in the current era of new media. Unfortunately, the Chinese version has not been updated to account for this limitation. Future studies should implement more sensitive assessment tools to validate the results of our study. Fourth, according to a meta-analytic study, the Montreal Cognitive Assessment (MoCA), as compared to the MMSE, better meets the criteria for screening and detecting mild cognitive impairment among individuals 60 years of age and older [37]. Future studies should consider implementing the MoCA to further examine cognitive health. In addition, our study excluded older adults who were unable to surf the Internet, suggesting that our results are not representative of all older adults. Finally, this study only included a small subset of the residents of Jinan city, and the eHealth literacy of the older adults is at a low level. Considering that eHealth literacy may be related to the Internet popularity and economic level [38], the results of this study may not be generalizable to other developed countries or districts. 

## 5. Conclusions

In summary, our study found that health-promoting lifestyles and eHealth literacy were each positively associated with cognitive health and the interaction of health-promoting lifestyles and eHealth literacy was negatively associated with cognitive health. In addition, eHealth literacy was associated with health-promoting lifestyles in older adults. Thus, interventions concerning healthy lifestyles and eHealth literacy should be provided to older adults to help improve their cognitive health. 

## Figures and Tables

**Table 1 ijerph-17-02263-t001:** Descriptive statistics for participant characteristics and differences in MMSE and HPLP by sociodemographic variables (*N* = 1201).

Variables	*n*	%	MMSE		*t/F* Value	HPLP		*t/F* Value
			Mean	*SD*		Mean	*SD*	
Age (in Years)					12.566 **			6.937 **
60–69	622	51.8	27.22	4.22		112.45	22.59	
70–79	485	40.4	27.30	4.51		113.58	24.28	
80 and above	94	7.8	24.87	5.24		103.90	20.46	
Gender					3.791 **			2.486 *
Male	560	46.6	27.59	3.86		114.01	23.89	
Female	641	53.4	26.61	4.90		110.68	22.58	
Residence					4.551 ***			6.784 ***
Urban area	320	26.6	28.03	3.94		119.65	25.21	
Rural area	881	73.4	26.72	4.60		109.54	21.89	
Education Level					16.217 **			36.416 **
Primary school and below	712	59.3	26.38	4.64		107.09	20.94	
Junior middle school	331	27.6	27.81	3.86		117.28	23.32	
High school	124	10.3	28.40	4.68		123.02	25.19	
University/college and above	34	2.8	29.44	1.80		131.47	27.60	
Marital Status					5.440 **			6.570 **
Married	884	73.6	27.48	4.09		114.83	23.38	
Other	317	26.4	25.91	5.23		105.00	21.29	
Family Economic Level					10.844 **			41.877 **
Low	340	28.3	26.14	4.79		103.05	22.03	
Medium	735	61.2	27.37	4.26		115.17	22.52	
High	126	10.5	27.80	4.41		119.87	23.55	

*Notes*: * *p* < 0.05; ***p* < 0.01; ****p* < 0.001. MMSE = Mini-Mental State Examination; HPLP = Health-Promoting Lifestyle Profile.

**Table 2 ijerph-17-02263-t002:** Correlations (*r*) between health-promoting lifestyle, eHealth literacy, and cognitive health (*N* = 1201).

Variables	HPLP	eHEALS	MMSE
HPLP	1		
eHEALS	0.454 ***	1	
MMSE	0.316 ***	0.227 ***	1

*Notes:* *** = *p* < 0.001. HPLP = Health-Promoting Lifestyle Profile; eHEALS = eHealth Literacy Scale; MMSE = Mini-Mental State Examination.

**Table 3 ijerph-17-02263-t003:** Associations of health-promoting lifestyle, eHealth literacy and cognitive health in Chinese older adults (*N* = 1201).

Predictors	Model 1 (MMSE)		Model 2 (MMSE)		Model 3 (HPLP)		Model 4 (MMSE)	
	*β*	*t* Value	*β*	*t* Value	*β*	*t* value	*β*	*t* Value
Age	−0.051	−1.824	−0.035	−1.209	0.042	1.620	−0.039	−1.398
Gender	−0.072	−2.609 **	−0.069	−2.462 *	−0.001	−0.039	−0.066	−2.393 *
Residence	−0.040	−1.337	−0.031	−0.994	−0.002	−0.061	−0.027	−0.877
Education level	0.072	2.288 *	0.087	2.709 **	0.105	3.582 ***	0.067	2.129 *
Marital status	−0.075	−2.633 **	−0.104	−3.609***	−0.121	−4.624 ***	−0.074	−2.617 **
Family economic level	0.016	0.561	0.038	1.281	0.106	3.931 ***	0.007	0.241
HPLP	0.262	9.038 ***					0.376	6.879 ***
eHEALS			0.152	4.949 ***	0.381	13.662 ***	0.443	3.377 **
HPLP×eHEALS							−0.465	−3.009 **
R^2^	0.128		0.087		0.250		0.137	
△R^2^	0.123		0.082		0.245		0.131	
*F*	25.038 ***		16.268 ***		56.698 ***		21.045 ***	

*Notes*: * = *p* < 0.05; ** = *p* < 0.01; *** = *p* < 0.001. HPLP = Health-Promoting Lifestyle Profile; eHEALS = eHealth Literacy Scale.

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
