# Peer review of "The Associations Among Health-Promoting Lifestyle, eHealth Literacy, and Cognitive Health in Older Chinese Adults: A Cross-Sectional Study"

_ijerph, 2020, doi:10.3390/ijerph17072263_

Round 1
Reviewer 1 Report
This paper presents a cross-sectional study examining the associations between a health-promoting lifestyle, eHealth literacy and cognitive health in Chinese older adults. A stratified cluster sampling method was used to collect data of 1201 older adults. My remarks are outlined below.
General comments:
The rationale to investigate whether the effect of having a health promoting lifestyle on cognitive health is moderated by eHealth literacy was not clear to me. Please provide more background information as well as a rationale in the Introduction-section. Furthermore, as the authors indicate that they examine the underlying mechanism of the relation between a healthy lifestyle and cognitive health, I expected a mediation rather than moderation analysis. Hypotheses 1 and 2 are checked using Pearson correlations. Consequently, there was no control for the effect of background variables (e.g. education level). It is not clear why a PROCESS macro was used to test the moderation effect. Why was a linear regression model not an option? Cognitive health will inevitably have an impact on eHealth literacy which might in turn affect health-promoting behaviours. This reverse pathway is not discussed in the introduction nor in the discussion.
Specific comments:
Line 72: "older adults are difficult to understand health knowledge". Please rephrase. Line 94: being able to surf the Internet using a mobile phone was an inclusion criteria. Why was it not sufficient that participants could surf the Internet using a computer? Line 154: 37.1% of participants were in good health. How was this measured? Line 175: The interaction effect of model 2 is discussed. However, in table 3 no interaction term is included in model 2. Line 2019: “people can suffer the Internet”. I believe this should be “people can surf the Internet”Author Response
Dear Reviewer,
Thank you very much for your careful review and constructive suggestions with regard to our manuscript " The Associations among Health-Promoting Lifestyle, eHealth Literacy and Cognitive Health in Chinese Older Adults:A Cross-Sectional Study "(ID: ijerph-677971). Those comments are very helpful for us to revise and improve our paper. We have studied comments carefully and have made correction which we hope meet with approval. A revised manuscript with the correction sections using the "Track Changes" function is attached to the supplemental material and for easy check/editing purpose. Our response to the comments is detailed in the document.

Reviewer 2 Report
This paper discusses three hypotheses. The first hypothesis relates to the association between health-promoting lifestyles and cognitive health, while the second hypothesis takes a more specific claim to the association between eHealth literacy and improvements in cognitive health. Similarly, the third hypothesis take a closer look at the role that eHealth literacy plays in moderating health-promoting lifestyles and cognitive health. A sample of 1201 older adults from distinct districts were provided with a questionnaire which found a positive link to support hypothesis 1 and 2. While hypothesis 3 had no clear results, it was discussed in-depth in both the results, discussion, and conclusion sections. The main contributions of this article are to the health of older adults and the overall health industry as it relates to analyzing possible methods/mechanisms that improve or better an aspect of geriatric health, in this case: cognitive health.
Major comments:
Introduction- The authors present the introduction in a very well organized and thoughtful way. The inclusion of both mild cognitive impairment and dementia references add to the substantiation of why this study is important. The description of health-promoting lifestyles was also a strength as it presents a clear description of the specific and overall factors to be analyzed further in the article. Within the introduction, the statement in lines 34-35 need review as cognitive impairment may not be the most “obvious” in the process of human aging. Other “obvious” factors in the process of human aging can include more physical aspects such as mobility, muscle weakness, etc. Please support your claim with citation or consider revising the wording. Throughout the introduction and in the abstract the concept of Quality of Life sticks out as imperative and lacks explanation within the body of the article. Consider adding factors that contribute to quality of life of older adults, or even what aspects of quality of life are affected by cognitive health. Methods- Within the methods section line 93 mentions “no serious diseases” as a description of the sample. Authors might consider expanding on this as it poses a conflict with the “seriousness” of cognitive health highlighted prior in the introduction. Cognitive impairment for instance might be considered a “serious disease” and it is unclear if that was measured prior to collecting information therefore, there is not true way to assess whether or not the participants in the sample had “no serious diseases”. Was there exclusion criteria for the sample? Study mentions that those who did not participate were simply not interested, but wonder if the research team had any exclusion criteria applied to sample other than age and city? There was informed consent from patients implied yet no IRB approval was mentioned or any methods to protect confidentiality or privacy were discussed. Was there code identifiers assigned rather than names collected? Was data stored securely and handled with privacy? In terms of measurements, the authors scales and measures were strongly evidence-based and user-friendly. Items in both the HPLP and the eHEALS were short and comprehensible. Consider adding a sample of these measures in your supplemental graphics. As an active clinical worker in a geriatric setting, there are many ways to measure cognitive health. While the MMSE is one of the most common ones, authors could consider using the Montreal Cognitive Assessment (MoCA) for future studies. A meta-analytic study presented data that MoCA testing as compared to the MMSE better meets criteria for screening tests for detection of mild cognitive impairment among those 60 years or older https://www.ncbi.nlm.nih.gov/pubmed/27992895. Another comparative study found that MoCA testing has less ceiling effect https://www.ncbi.nlm.nih.gov/pmc/articles/PMC4562190/. Results- Well organized and presented. Graph added to the quality of the presentation of the results. Chart was easy to comprehend and interpret. It would be helpful to possibly include a key or brief description of what is considered a normal cognitive score versus an abnormal one. It is significant to note normal aging cognitive health scores versus possible MCI or dementia scores. Discussion- A strong discussion was presented and limitations were clearly explained. I wonder if the setting/environment in which the survey was conducted was a limitation to the sample or overall study?
Minor comments:
Health status mentioned in abstract and not explained in article. Line 38 of introduction consider re-wording as “huge burden of disease” is unclear. What is burden of disease? Might consider something such as “brings a major prevalence of disease”. Alternatively, authors could explain “burden of disease” In other words, highlight or specify how poor cognitive health is a burden of disease in the context of global aging. Line 71-72 also needs re-wording as it reads “older adults are difficult to understand health knowledge…” It is unclear of what the authors meant to convey in this statement. Consider including more citations throughout the paper specifically related to Chinese older adults and health-promoting lifestyles as it would add to the originality and contribution of this paper to the greater society. It is so important to highlight minority population data and diversity.Author Response
Dear Reviewer,
Thank you very much for your careful review and constructive suggestions with regard to our manuscript " The Associations among Health-Promoting Lifestyle, eHealth Literacy and Cognitive Health in Chinese Older Adults:A Cross-Sectional Study "(ID: ijerph-677971). Those comments are very helpful for us to revise and improve our paper. We have studied comments carefully and have made correction which we hope meet with approval. A revised manuscript with the correction sections using the "Track Changes" function is attached to the supplemental material and for easy check/editing purpose. Our response to the comments is detailed in the document.

Reviewer 3 Report
The content of this research topic is interesting and it is very meaningful to reconfirm the factors related to health promotion in an aging society.
However, there is a need to improve overall readability by organizing general English sentences.
In particular, it is necessary to explain the method of research more concisely and reasonably describe the sample size.
The research results need to explain the results of research on the hypothesis in detail. In addition, further information is needed regarding the future application of this study.
Author Response
Dear Reviewer,
Thank you very much for your careful review and constructive suggestions with regard to our manuscript " The Associations among Health-Promoting Lifestyle, eHealth Literacy and Cognitive Health in Chinese Older Adults:A Cross-Sectional Study "(ID: ijerph-677971). Those comments are very helpful for us to revise and improve our paper. We have studied comments carefully and have made correction which we hope meet with approval. A revised manuscript with the correction sections using the "Track Changes" function is attached to the supplemental material and for easy check/editing purpose. Our response to the comments is detailed in the document.

Round 2
Reviewer 1 Report
Dear authors, thank you for your adaptations to the manuscript. However, in my opinion, these alterations do not create the clarity I was missing in the first version of the paper. I was still left with a number of important questions after reading the manuscript. I have outlined these questions/remarks below.
Based on your introduction it remains unclear why eHealth literacy would moderate or mediate the relation between having a health-promoting lifestyle and cognitive health. To be more specific, I don’t see how the two sentences “Indeed, previous research indicates that eHealth literacy mediated the effect of individual factors on health behaviors [31]” (line 87-89) and “Thus, we infer that a two-way relationship between health-promoting lifestyle and cognitive health may be affected by individual eHealth literacy.” (line 89-90) are linked. In the previous feedback round I indicated that the reverse pathway was also possible. I thought you would acknowledge that the data are cross-sectional and discuss the follow-up research needed to determine the direction of the effects. You added the reverse pathway as a new hypothesis. However, in my opinion, this makes the paper more difficult to the reader. A sample size calculation was added. However, this calculation is not clear to me. I would recommend to add a reference for the formula and explain why you used the prevalence of mild cognitive impairment in older Chinese adults to calculate the sample size. Was this sample size calculated before the start of the study? line 197-198: “Model 2 suggested that eHealth literacy positively predicted cognitive health(β = 0.135, p < 0.01), supporting hypothesis 2.” Model 2 also shows a significant interaction effect between eHealth literacy and a health-promoting lifestyle. Consequently, the main effect can no longer be interpreted. Furthermore, when looking at model 1 no significant main effect of eHealth literacy is found. Multilevel analyses might be more appropriate given that people belonged to clusters (i.e. urban communities or natural villages). The mediation analyses are unclear.Author Response
Dear Reviewer,
Thank you very much for your careful review and constructive suggestions with regard to our manuscript " The Associations among Health-Promoting Lifestyle, eHealth Literacy and Cognitive Health in Chinese Older Adults:A Cross-Sectional Study "(ID: ijerph-677971) again. Those comments are very helpful for us to revise and improve our paper. We have studied comments carefully and have made correction which we hope meet with approval. A revised manuscript with the correction sections using the "Track Changes" function is attached to the supplemental material and for easy check/editing purpose. Our response to the comments is as follows:
Point 1: Based on your introduction it remains unclear why eHealth literacy would moderate or mediate the relation between having a health-promoting lifestyle and cognitive health. To be more specific, I don’t see how the two sentences “Indeed, previous research indicates that eHealth literacy mediated the effect of individual factors on health behaviors [31]” (line 87-89) and “Thus, we infer that a two-way relationship between health-promoting lifestyle and cognitive health may be affected by individual eHealth literacy.” (line 89-90) are linked. In the previous feedback round I indicated that the reverse pathway was also possible. I thought you would acknowledge that the data are cross-sectional and discuss the follow-up research needed to determine the direction of the effects. You added the reverse pathway as a new hypothesis. However, in my opinion, this makes the paper more difficult to the reader.
Response 1: Thank you for your comments. First of all, I'm sorry that I misunderstood you. So we delete the mediation analysis. And we briefly describe the reverse path in the limitations section In addition, after carefully considering your opinion, we searched some literature, but didn't find the theoretical basis of eHealth literacy moderation. Therefore, we cannot explain the theoretical basis of eHealth literacy moderate the impact of health-promoting lifestyle on cognitive health. So we put it in a different way, and we removed the moderating effect diagram. We changed to study the impact of the interaction between health-promoting lifestyle and eHealth literacy on cognitive health. We don't know whether this modification is appropriate, it's the best way we can think of. We hope we can get your approval.
Point 2: A sample size calculation was added. However, this calculation is not clear to me. I would recommend to add a reference for the formula and explain why you used the prevalence of mild cognitive impairment in older Chinese adults to calculate the sample size. Was this sample size calculated before the start of the study?
Response 2: Thank you for your careful checks. In the revised version, we added a reference. The original purpose of this study was to study the status and influencing factors of mild cognitive impairment in the elderly, so we used the prevalence of mild cognitive impairment to calculate the sample size. And this sample size was calculated before the start of the study.
Point 3: line 197-198: “Model 2 suggested that eHealth literacy positively predicted cognitive health(β = 0.135, p < 0.01), supporting hypothesis 2.” Model 2 also shows a significant interaction effect between eHealth literacy and a health-promoting lifestyle. Consequently, the main effect can no longer be interpreted. Furthermore, when looking at model 1 no significant main effect of eHealth literacy is found.
Response 3: Thank you for your careful checks. I am sorry that it was caused by my carelessness. In the revised version, we revised the data.
|
Predictors |
Model 1(MMSE) |
Model 2 (MMSE) |
Model 3 (HPLP) |
Model 4 (MMSE) |
||||
|
β |
t value |
β |
t value |
β |
t value |
β |
t value |
|
|
Age |
-0.051 |
-1.824 |
-0.035 |
-1.209 |
0.042 |
1.620 |
-0.039 |
-1.398 |
|
Gender |
-0.072 |
-2.609** |
-0.069 |
-2.462* |
-0.001 |
-0.039 |
-0.066 |
-2.393* |
|
Residence |
-0.040 |
-1.337 |
-0.031 |
-0.994 |
-0.002 |
-0.061 |
-0.027 |
-0.877 |
|
Education level |
0.072 |
2.288* |
0.087 |
2.709** |
0.105 |
3.582*** |
0.067 |
2.129* |
|
Marital status |
-0.075 |
-2.633** |
-0.104 |
-3.609*** |
-0.121 |
-4.624*** |
-0.074 |
-2.617** |
|
Family economic level |
0.016 |
0.561 |
0.038 |
1.281 |
0.106 |
3.931*** |
0.007 |
0.241 |
|
HPLP |
0.262 |
9.038*** |
|
|
|
|
0.376 |
6.879*** |
|
eHEALS |
|
|
0.152 |
4.949*** |
0.381 |
13.662*** |
0.443 |
3.377** |
|
HPLP×eHEALS |
|
|
|
|
|
|
-0.465 |
-3.009** |
|
R2 |
0.128 |
0.087 |
0.250 |
0.137 |
||||
|
â–³R2 |
0.123 |
0.082 |
0.245 |
0.131 |
||||
|
F |
25.038*** |
16.268*** |
56.698*** |
21.045*** |
||||
Point 4: Multilevel analyses might be more appropriate given that people belonged to clusters (i.e. urban communities or natural villages).
Response 4: Thank you for your comments. In the revised version, we included residence in the single factor analysis, and it is taken as the control variable in the regression analysis. We didn't do the multilevel analyses because residence had no effect on cognitive health or health-promoting lifestyle in the regression analysis
Point 5: The mediation analyses are unclear.
Response 5: Thank you for your careful checks. In the revised version, I have deleted it.

Round 3
Reviewer 1 Report
Dear authors,
thank you for your adaptations to the paper. However, I believe that there are still a number of important issues with the paper.
- abstract: "an interaction of health-promoting lifestyle and eHealth literacy positively predicted cognitive health". However, the beta shows a negative value. In the discussion you indicate that the interaction term negatively predicts cognitive health. I do not believe that this significant interaction term is explained in a clear way to the reader. A negative interaction term does not mean that the 2 predictors have "some of the same effects on cognitive health".
- Please explain hypothesis 4. Which interaction do you expect to find and why?
- Why was eHealth literacy entered as a binary variable to the model? In general it is advised not to dichotomize continuous variables (for example see Babyak, M. A. (2004). What you see may not be what you get: a brief, nontechnical introduction to overfitting in regression-type models. Psychosomatic medicine, 66(3), 411-421.)
- During the previous review round I indicated that multilevel analyses might be more suited given that you used a stratified cluster sampling method. Now you only account for whether or not people live in a rural or urban area. However, participants living in the same street will be more similar than participants living in different streets. By conducting multilevel analyses the clustering in the data will be taken into account.
- Only 11% of your sample had an adequate eHealth literacy. How will this have affected the results of the analysis?
Author Response
Dear Reviewer,
Thank you very much for your careful review and constructive suggestions with regard to our manuscript. Those comments are very helpful for us to revise and improve our paper. We have studied comments carefully and have made correction which we hope meet with approval. Our detailed response to the comments is as follows:
Point 1: abstract: "an interaction of health-promoting lifestyle and eHealth literacy positively predicted cognitive health". However, the beta shows a negative value. In the discussion you indicate that the interaction term negatively predicts cognitive health. I do not believe that this significant interaction term is explained in a clear way to the reader. A negative interaction term does not mean that the 2 predictors have "some of the same effects on cognitive health".
Response 1: Thank you for your comments. I do apologize for this confusion – it was caused by my carelessness. In the revised version of the manuscript, I have changed the abstract and the discussion.
Point 2: Please explain hypothesis 4. Which interaction do you expect to find and why?
Response 2: Thank you for your careful checks. In the revised version, we discuss the expected interaction.
Point 3: Why was eHealth literacy entered as a binary variable to the model? In general it is advised not to dichotomize continuous variables (for example see Babyak, M. A. (2004). What you see may not be what you get: a brief, nontechnical introduction to overfitting in regression-type models. Psychosomatic medicine, 66(3), 411-421.)
Response 3: Thank you for your reminding. In fact, we used continuous variables for analysis in the previous version, but forgot to delete the description of binary variables. I apologize again for my carelessness. In the revised version, the relevant description of binary variables has been removed to avoid ambiguity.
Point 4: During the previous review round I indicated that multilevel analyses might be more suited given that you used a stratified cluster sampling method. Now you only account for whether or not people live in a rural or urban area. However, participants living in the same street will be more similar than participants living in different streets. By conducting multilevel analyses the clustering in the data will be taken into account.
Response 4: Thank you for your comments. Unfortunately, we could not conduct multilevel analyses because of the lack of street information. Due to insufficient consideration before the survey, we only collected the rural or urban location of residence and did not record the street information.
Point 5: Only 11% of your sample had an adequate eHealth literacy. How will this have affected the results of the analysis?
Response 5: Thank you for your careful checks. This may affect the extrapolation of the study, so I've made this point in the limitations section.
Thank you very much for your painstaking review again.